
# The inheritance of energy conditions: Revisiting no-go theorems in string compactifications

Heliudson Bernardo[1⋆], Suddhasattwa Brahma[2†] and Mir Mehedi Faruk[1,2,3,4‡]

**1** Department of Physics, McGill University, Montreal, QC, H3A 2T8, Canada
**2** Higgs Centre for Theoretical Physics, School of Physics & Astronomy,
University of Edinburgh, Edinburgh EH9 3FD, UK
**3** Institute of Physics, University of Amsterdam, Science Park 904,
PO BOX 94485, 1090 GL Amsterdam, Netherlands
**4** Delta Institute for Theoretical Physics, Science Park 904,
PO Box 94485, 1090 GL Amsterdam, Netherlands

⋆ heliudson@hep.physics.mcgill.ca , † suddhasattwa.brahma@gmail.com ,
‡ mir.faruk@mail.mcgill.ca

## Abstract

One of the fundamental challenges in string theory is to derive realistic four-dimensional cosmological backgrounds from it despite strict consistency conditions that constrain its possible low-energy backgrounds. In this work, we focus on energy conditions as *covariant and background-independent* consistency requirements in order to classify possible backgrounds coming from low-energy string theory in two steps. Firstly, we show how supergravity actions obey many relevant energy conditions under some reasonable assumptions. Remarkably, we find that the energy conditions are satisfied even in the presence of objects which individually violate them due to the tadpole cancellation condition. Thereafter, we list a set of conditions for a higher-dimensional energy condition to imply the corresponding lower-dimensional one, thereby categorizing the allowed low-energy solutions. As for any no-go theorem, our aim is to highlight the assumptions that must be circumvented for deriving four-dimensional spacetimes that necessarily violate these energy conditions, with emphasis on cosmological backgrounds.

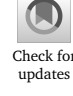

# 1 Introduction

In Einstein's general relativity (GR), spacetimes are manifolds endowed with dynamical geometric properties such that certain energy sources produce singularities. This is best seen from requirements of geodesic completeness, which includes conditions for the energy-momentum tensor defined on the manifold [1]. The null and strong energy conditions (NEC and SEC, respectively) are crucial for the Hawking-Penrose singularity theorems [2–4] which, roughly speaking, state that singularities in GR sourced by physical matter are unavoidable.

It is expected that a quantum version of the theory would resolve spacetime singularities, and a preeminent candidate for quantum gravity is string theory. Despite the elusive absence of a non-perturbative and background-independent formulation, string theory provides new theoretical insights about the nature of gravity (*e.g.* holography) and fields. Even at the classical level, it modifies GR by adding higher curvature corrections to the Einstein-Hilbert action, and its consistency constrains the matter-energy content that sources the corrected Einstein's equations.

The low-energy limit of weakly coupled string theory is described by higher-dimensional supergravity theories and any consistent background for the theory should, in that limit, be a solution to specific supergravity equations. It is then natural to ask which spacetimes can be obtained from such a theory. This is restricted by the matter-energy content that can be consistently evoked. In fact, in the context of compactifications to four-dimensions, there are no-go theorems preventing the solutions to exhibit a four-dimensional manifold with positive curvature. In [5], this was shown after some reasonable assumptions compatible with the low-energy string actions (see also [6]), and in [7] this was extended to include D-branes and orientifold planes.

Another approach to sort out which background solutions string theory might give is to study energy conditions that can be satisfied, given the fields present in the theory. This is important because physically relevant backgrounds are tied to relevant energy conditions, such as the SEC (which must be violated in a four-dimensional spatially flat and accelerating FLRW spacetime) and NEC (which has to be saturated for a de Sitter (dS) spacetime). In particular, it was shown in [8] that a non-singular, time-dependent compactification might give a $dS_4$ space without violating the higher-dimensional SEC, but that the SEC and NEC together prevents such a solution. Indeed, time-dependent compactifications were also shown to avoid no-go theorems in [9] and [10–12], although curvature and quantum corrections were also present in the latter, and the interplay of this work with the NEC was studied in [13], where it was found out that the lower-dimensional NEC is necessary for a higher-dimensional effective field theory description of the full background solution in M-theory. There are more recent results in understanding string compactifications and its relation to cosmological so-

lutions (see [14–18] and references therein). For instance, in [19] detailed no-go theorems are presented when reducing non-supersymmetric string actions on compact manifolds, preventing the lower-dimensional theory from having a positive cosmological constant. Similar conclusions are obtained in the study of higher-dimensional Raychaudhuri equation [20] and worldsheet analysis [21]. It was discussed in [22] that the Virasoro constraints of the worldsheet theory imply precisely the geometric form of the NEC.[1] Yet further difficulties in obtaining accelerating cosmological solutions are reported in [25] and some other related works on time-dependent compactifications [26–28]. Although the internal compact spacetimes taken in these works are Ricci flat or conformally Ricci flat, yet this is a motivation to study general time-dependent internal manifolds when considering the violation of energy conditions in string compactifications.

The gist of the Gibbons-Maldacena-Nunez (GMN) no-go theorem can be reformulated as follows: The assumptions of the theorem are sufficient to guarantee the validity of the strong energy condition in four dimensions [29]. Motivated by this fact, in this paper, we investigate the requirements for certain energy conditions to still hold after the dimensional reduction procedure, that is, we study under which conditions certain $D$-dimensional energy conditions imply the lower-dimensional ones. Specifically, we study the null, strong, weak, and dominant energy conditions. As we will discuss, given that string supergravity actions satisfy all the above energy conditions for any physical field configuration, and if the compactification satisfies some modest requirements, then it follows that the lower-dimensional energy conditions will necessarily be satisfied and, more explicitly, no solution which violates them is realizable. This new form of no-go theorems, offered by considering energy conditions, is very powerful and, as with any such result, only limited by the assumptions which go into deriving them. Identifying these restrictions will indicate how to come up with string effects that systematically circumvent them, and yield realistic low-energy backgrounds, which will be particularly useful for cosmological applications.

In the next section, we revisit the energy conditions for the field content of supergravity, with a main emphasis on $p$-form fields, showing that such fields satisfy the null, strong, weak, and dominant energy conditions for any field configuration. We also comment on how extended sources like D-branes and orientifold planes satisfy them, showing that the tadpole cancellation condition is *crucial* for the validity of the energy conditions for systems containing the latter. In section 3, we find sufficient requirements for $D$-dimensional energy conditions to imply the lower $d$-dimensional ones. This is the main result of this paper and provides another explanation of why it is so difficult to obtain certain four-dimensional spacetimes in low-energy string theory. In section 4 we summarize and discuss the implications of our results.

**Notation and conventions:** Mostly plus signature is used throughout this work and Planckian units are employed when not stated otherwise. In many instances, we use an overbar to stress that certain components are computed for a $D$-dimensional metric and we assume $D > 2$. The adopted convention for the Riemann curvature tensor components is $\overline{R}_{MNP}{}^{Q} = -2\partial_{[M}\overline{\Gamma}_{N]P}^{Q} + 2\overline{\Gamma}_{P[M}^{L}\overline{\Gamma}_{N]L}^{Q}$.

## 2 Energy conditions for fields and sources

Before discussing energy conditions for each individual class of fields, let us briefly summarize the physical meaning and quantitative formulation of the conditions we consider in this paper.

---

[1]Conditions on the internal components of the energy-momentum tensor of supergravity theories have been studied in [23, 24], where constraints on the internal geometry were discussed.

Denoting $R_{MN}$ and $R$ the Ricci tensor and scalar, respectively, of a spacetime with metric $g_{MN}$, we have

- *Null energy condition (NEC):* Any congruence of null geodesics is everywhere non-divergent. In particular, curvature cannot make parallel lightrays diverge from each other. Quantitatively,

$$R_{MN}(x)l^M l^N \geq 0, \qquad \forall \quad l^M(x) \, : \, g_{MN}(x)l^M l^N = 0. \qquad (1)$$

- *Strong energy condition (SEC):* Any congruence of timelike geodesics is everywhere non-divergent. Physically, this is essentially the fact that gravity is non-repulsive,

$$R_{MN}(x)u^M u^N \geq 0, \qquad \forall \quad u^M(x) \, : \, g_{MN}(x)u^M u^N < 0. \qquad (2)$$

- *Weak energy condition (WEC):* Locally, any observer perceives non-negative energy density and non-repulsive gravity. Mathematically,

$$\left(R_{MN}(x) - \frac{1}{2}R(x)g_{MN}(x)\right)u^M u^N \geq 0, \quad \forall \quad u^M(x) \, : \, g_{MN}(x)u^M u^N < 0. \qquad (3)$$

- *Dominant energy condition (DEC):* Locally, any observer measures a causal flow of non-negative energy and non-repulsive gravity. On top of the WEC, the flux condition for the Einstein tensor $G_{MN}$,

$$g_{MN}(x)G^M{}_P(x)G^N{}_Q(x)u^P u^Q \leq 0, \qquad (4)$$

should be satisfied everywhere for any timelike vector field $u^M(x)$.

One important aspect of these conditions is that they are background-independent, *i.e.* valid for any metric. It is then of no surprise that we might find background-independent conditions for the possible four-dimensional spacetimes coming from string compactifications by reformulating the no-go theorems in terms of their validity. For instance, instead of asking whether or not we can obtain a $dS_4$ at low energies, one might look for conditions of SEC violation, which is a metric-independent question that, if answered, might also shed light on the fate of other cosmological backgrounds in string theory.

An important result that we will make use of in this section is the following: *any inequality $f(u_1, u_2) \geq 0$ involving timelike vectors fields $u_1^M$ and $u_2^M$ and a continuous function $f(x, y)$ implies also the related inequality $f(l_1, l_2) \geq 0$ for null vectors fields $l_1^M$ and $l_2^M$.* This is shown in [30] and also discussed in [31, 32] which we refer to more detailed discussions on the significance of energy conditions (see also [33, 34]). Roughly speaking, it follows from a limiting procedure on choosing timelike vectors which have very small modulus. This result is the reason why the SEC implies the NEC and the WEC implies the NEC. Moreover, by definition, the DEC implies the WEC. For completeness, we shall study each energy condition individually.

Note that we have stated geometric conditions without mentioning any restrictions on energy-momentum tensors. When Einstein's equations are satisfied, there is no difference between these two approaches, but this is not the case once we depart from a second-derivative action for the metric. When applying the results of this paper to string theory, we assume low-energy second-derivative supergravity actions, so the geometric energy conditions are enough. Notwithstanding this caveat, the idea of constraining lower-dimensional backgrounds using higher-dimensional geometric conditions might still be worth studying even after adding curvature corrections, with the proviso that we should put all corrections in the "matter" sector

and consider an effective energy-momentum tensor. However, the specifics will change, and we prefer to focus on the supergravity limit for the rest of this work.

Let us proceed with the energy conditions for the matter content which appear in the (bosonic sector of) supergravity actions, including low-energy M-theory. Schematically, they have the form,

$$S = \frac{1}{2\kappa_D^2} \int d^D x \sqrt{-g} \left[ R - \frac{1}{2} \partial_\mu \phi \, \partial^\mu \phi - \frac{\kappa_D^2}{2g_D^2} \sum_p e^{\beta_p \phi} |F_p|^2 \right] + \sum_p \int A_p \wedge \pi_{D-p} + S_{\text{local sources}},$$
(5)

where $F_p$ is not necessarily the field strength of $A_p$ but might also include couplings with other fields and $\pi_{D-p}$ contain sources and terms that resemble Chern-Simons (CS) couplings to different $p$-form fields. We have also written a possible dilaton kinetic couplings $\beta_p$ to the $p$-form fields.

In the rest of this section, we show that actions of the form above satisfy all the previous energy conditions for any field configuration. The cases of a 1-form field and scalar fields with arbitrary potentials are well-known in the literature (e.g. see [31,35]), the DEC and SEC are satisfied for the former while, for the latter, the validity of the conditions depends on the potential: for $V \geq 0$, DEC is satisfied but SEC might be violated when $V \neq 0$ while for $V \leq 0$ SEC is respected but DEC might not hold when $V \neq 0$. That is one of the reasons why there is a link between the GMN no-go theorem (which assumes a non-positive potential for the dilaton – except for the massive type IIA case) and the strong energy condition validity. In the following, we focus on the other fields and sources that appear in supergravity actions.

## 2.1 $p$-form fields

Consider $p$-form fields $A_p$ with action

$$S[A_p] = -\frac{1}{2} \int e^{\lambda \phi} * F_{p+1} \wedge F_{p+1}$$
$$= -\frac{1}{2} \frac{1}{(p+1)!} \int d^D x \sqrt{-g} \, e^{\lambda \phi} g^{M_1 N_1} \cdots g^{M_{p+1} N_{p+1}} F_{M_1 \cdots M_{p+1}} F_{N_1 \cdots N_{p+1}}.$$
(6)

We have included the possibility of a kinetic coupling to a scalar field $\phi$. As discussed before, scalar fields might violate some energy condition depending on their potential, which is highly constrained in ten-dimensional supergravities coming from string theory. Recall that since the Bianchi identity for $F_{p+1}$ allows us to define a dual theory for a $(D-p-2)$-form, it suffices us to consider $p < D-2$. Note also that the case $p = 0$ might be treated as a scalar field without potential, for which all energy conditions are respected.

The energy-momentum tensor for $S[A_p]$ is given by

$$T_{MN}^{(p)} = e^{\lambda \phi} \left( \frac{1}{p!} F^{M_1 \cdots M_p}{}_M F_{M_1 \cdots M_p N} - \frac{1}{2(p+1)!} g_{MN} F^{M_1 \cdots M_{p+1}} F_{M_1 \cdots M_{p+1}} \right).$$
(7)

Evidently, the value of $\lambda$ plays no role in whether $S[A_p]$ satisfies a given energy condition or not. So, to avoid clutter, we set $\lambda = 0$ for the rest of this section.

To check whether WEC is respected or not, let us contract with an arbitrary timelike vector $u^M$ to get $T_{MN}^{(p)} u^M u^N$. Going to a local vielbein basis, $e_M^A(x)$, we can write $T_{AB}^{(p)} u^A u^B$ where capital letters from the earlier part of the Latin alphabet denote local (flat) indices, *i.e.* $g^{MN} e_M^A e_N^B = \eta^{AB}$ where $\eta_{AB}$ is the $D$-dimensional Minkowski metric. Using a Lorentz transformation, we can always go to a frame on which $u^A = (u^{\bar{0}}, 0, \ldots, 0)$, where the overbar is a

reminder that the index is a flat one. In this case

$$
\begin{aligned}
T^{(p)}_{MN}u^M u^N &= T^{(p)}_{AB}u^A u^B \\
&= \left(u^{\bar 0}\right)^2 \left(\frac{1}{p!}F^{I_1\cdots I_p}{}_{\bar 0}F_{I_1\cdots I_p\bar 0} + \frac{1}{2(p+1)!}F^{A_1\cdots A_{p+1}}F_{A_1\cdots A_{p+1}}\right),
\end{aligned}
\tag{8}
$$

where all indices appearing in the last line are local indices. Writing the last term in the second line as

$$
F^{A_1\cdots A_{p+1}}F_{A_1\cdots A_{p+1}} = -(p+1)F^{I_1\cdots I_p}{}_{\bar 0}F_{I_1\cdots I_p\bar 0} + F^{I_i\cdots I_{p+1}}F_{I_i\cdots I_{p+1}},
\tag{9}
$$

we get

$$
T^{(p)}_{AB}u^A u^B = \left(u^{\bar 0}\right)^2 \left(\frac{1}{2p!}\,\delta^{I_1 J_1}\cdots\delta^{I_p J_p}F_{I_1\cdots I_p\bar 0}F_{J_1\cdots J_p\bar 0} + \frac{1}{2(p+1)!}\delta^{I_1 J_1}\cdots\delta^{I_{p+1}J_{p+1}}F_{I_1\cdots I_{p+1}}F_{J_1\cdots J_{p+1}}\right),
\tag{10}
$$

which is explicitly greater than or equal to zero. Hence, the WEC energy condition is satisfied. By continuity, this also shows that NEC is satisfied since the contraction above is a smooth function of the timelike vector $u^M$.

To check for the validity of the SEC, we need the trace of $T^{(p)}_{MN}$:

$$
T^{(p)} := T^{(p)}_{MN}g^{MN} = T^{(p)}_{AB}\eta^{AB} = \frac{D}{2(p+1)!}\left(\frac{2}{D}(p+1)-1\right)F^{A_1\cdots A_{p+1}}F_{A_1\cdots A_{p+1}}.
\tag{11}
$$

Using (9) in the SEC inequality and assuming $u^A$ has vanishing space components, we have

$$
\left(T^{(p)}_{MN} - \frac{1}{D-2}T^{(p)}g_{MN}\right)u^M u^N = \frac{\left(u^{\bar 0}\right)^2}{2(p+1)!}\frac{1}{D-2}\Big[2(p+1)(D-p-2)F^{I_1\cdots I_p}{}_{\bar 0}F_{I_1\cdots I_p\bar 0} \\
+ 2pF^{I_1\cdots I_{p+1}}F_{I_1\cdots I_{p+1}}\Big].
\tag{12}
$$

For $p < D-2$, the contraction above is greater than or equal to zero and so SEC is satisfied.

Finally, for the DEC, we need to check whether $V^M := -T^{(p)M}{}_N u^N$ is a causal vector or not. To accomplish that, we shall work with the electric and magnetic form fields, with components defined as

$$
E_{M_1\cdots M_p} = \frac{1}{p!}F_{M_1\cdots M_p N}u^N,
\tag{13}
$$

$$
B_{M_1\cdots M_{D-(p+2)}} = \frac{1}{(p+1)!}\epsilon_{M_1\cdots M_{D-(p+2)}N_1\cdots N_{p+1}Q}u^Q F^{N_1\cdots N_{p+1}},
\tag{14}
$$

where $\epsilon_{M_1\cdots M_D}$ is the Levi-Civita tensor. These generalize the $p=1$ electric and magnetic fields as measured by the congruence of observers defined by $u^N$. Note that the electric and magnetic form fields are both transverse to $u^N$, a property that we shall use in this section. In terms of $E$ and $B$, we can write the field strength as

$$
F_{M_1\cdots M_{p+1}} = (p+1)!(-1)^{p+1}u_{[M_1}E_{M_2\cdots M_{p+1}]} + \frac{(-1)^{(p+1)(D-(p+2))}}{(D-(p+2))!}\epsilon_{M_1\cdots M_{p+1}N_1\cdots N_{D-(p+2)}Q}u^Q \\
\times B^{N_1\cdots N_{D-(p+2)}}.
\tag{15}
$$

Plugging this expression in the energy-momentum tensor and after some algebraic manipulations, we get

$$T_{MN} = p! \left( p u^2 E_M{}^{M_2 \cdots M_p} E_{N M_2 \cdots M_p} + u_M u_N E^2 \right) - \frac{g_{MN}}{2} \left[ p! u^2 E^2 - \frac{1}{(D-(p+2))!} u^2 B^2 \right]$$
$$+ 2 \frac{(-1)^{(p+1)(D-(p+1))}}{(D-(p+2))!} u_{(M} \epsilon_{N) M_1 \cdots M_p L_1 \cdots L_{D-(p+2)} Q} u^Q B^{L_1 \cdots L_{D-(p+2)}} E^{M_1 \cdots M_p}$$
$$- \frac{1}{(D-(p+2))!} \left[ g_{MN} u^2 B^2 - u_M u_N B^2 - (D-(p+2)) u^2 B_{M N_2 \cdots N_{D-(p+2)}} B_N{}^{N_1 \cdots N_{D-(p+2)}} \right], \tag{16}$$

where we defined

$$E^2 = E_{M_1 \cdots M_p} E^{M_1 \cdots M_p}, \qquad B^2 = B_{N_1 \cdots N_{D-(p+2)}} B^{N_1 \cdots N_{D-(p+2)}}. \tag{17}$$

Hence, the energy-flux vector can be written as

$$V_M = -T_{MN} u^N$$
$$= -u^2 \frac{u_M}{2} \left( p! E^2 + \frac{1}{(D-(p+2))!} B^2 \right) - \frac{(-1)^{(p+1)(D-(p+1))}}{(D-(p+2))!} u^2 \epsilon_{M M_1 \cdots M_p N_1 \cdots N_{D-(p+2)} Q} u^Q$$
$$\times B^{N_1 \cdots N_{D-(p+2)}} E^{M_1 \cdots M_p}. \tag{18}$$

The first term is proportional to the energy-density flux, while the second term comes from a generalization of the Umov-Poynting vector usually discussed in the $p = 1$ case. To check whether $V^M$ is causal or not, we need to compute $V^2 = V^M V_M$:

$$V^2 = (u^2)^2 \left[ \frac{u^2}{4} \left( p! E^2 + \frac{B^2}{(D-(p+2))!} \right)^2 + \frac{u^Q u_S}{((D-(p+2))!)^2} \epsilon_{M M_1 \cdots M_p N_1 \cdots N_{D-(p+2)} Q} \right.$$
$$\left. \times \epsilon^{M J_1 \cdots J_p L_1 \cdots L_{D-(p+2)} S} B^{N_1 \cdots N_{D-(p+2)}} E^{M_1 \cdots M_p} B_{L_1 \cdots L_{D-(p+2)}} E_{J_1 \cdots J_p} \right]. \tag{19}$$

To simplify the second term in the square bracket, we use,

$$\epsilon_{M M_1 \cdots M_p N_1 \cdots N_{D-(p+2)} Q} \epsilon^{M J_1 \cdots J_p L_1 \cdots L_{D-(p+2)} S} = -(D-1)! \delta_{M_1}^{[J_1} \cdots \delta_{M_p}^{J_p} \delta_{N_1}^{L_1} \cdots \delta_{N_{D-(p+2)}}^{L_{D-(p+2)}} \delta_Q^{S]}. \tag{20}$$

Since any contraction of $u^M$ with the components of the electric and magnetic field forms vanishes, the only terms in the antisymmetrization above that will contribute to $V^2$ are the ones proportional to $\delta_Q^S$, that is,

$$\epsilon_{M M_1 \cdots M_p N_1 \cdots N_{D-(p+2)} Q} \epsilon^{M J_1 \cdots J_p L_1 \cdots L_{D-(p+2)} S} u^Q u_S B^{N_1 \cdots N_{D-(p+2)}} E^{M_1 \cdots M_p} B_{L_1 \cdots L_{D-(p+2)}} E_{J_1 \cdots J_p}$$
$$= -(D-2)! u^2 E^{[J_1 \cdots J_p} B^{L_1 \cdots L_{D-(p+2)}]} E_{J_1 \cdots J_p} B_{L_1 \cdots L_{D-(p+2)}}. \tag{21}$$

We will separate the antisymmetric factor above into terms that include only the antisymmetrization of the $J$ and $L$ indices separately and terms that include a mixed exchange of $J$ and $L$ indices. There are $C_r^p C_r^{D-(p+2)} p! (D-(p+2))!$ terms containing $r$ exchanges of the $J$ and $L$ indices (number of ways of combining $r$ $L$ indices into $p$ possible slots times number of ways of combining $r$ $J$ indices into $D-(p+2)$ slots times the individual antisymmetrization of $J$ and $L$ indices), so

$$-(D-2)! u^2 E^{[J_1 \cdots J_p} B^{L_1 \cdots L_{D-(p+2)}]} E_{J_1 \cdots J_p} B_{L_1 \cdots L_{D-(p+2)}} = -u^2 p! (D-(p+2))! \left[ E^2 B^2 \right.$$
$$\left. + \sum_{r=1}^{r_{max}} (-1)^r C_r^p C_r^{D-(p+2)} (E \cdot B)^{L_2 \cdots L_r J_{r+1} \cdots J_p}_{L_2 \cdots L_{D-(p+2)}} (E \cdot B)_{J_2 \cdots J_p}^{J_2 \cdots J_r L_{r+1} \cdots L_{D-(p+2)}} \right], \tag{22}$$

where $r_{\max} = \min(p, D-(p+2))$ and we denoted

$$(E \cdot B)^{J_2 \cdots J_p}{}_{L_2 \cdots L_{D-(p+2)}} = E^{J_1 \cdots J_p} B_{L_1 \cdots L_{D-(p+2)}} \delta^{L_1}_{J_1} . \tag{23}$$

Due to the antisymmetrization, terms with an even number $r = 2n$ of exchanges positively contribute to $V^2$ (since we assume $u^2 < 0$). However, such terms will be combined with some components of the $r = 2n-1$ terms. To see that, consider the sum of two such successive terms,

$$(-1)^{2n-1} c_{2n-1} (E \cdot B)^{L_2 \cdots L_{2n-1} J_{2n} \cdots J_p}{}_{L_2 \cdots L_{D-(p+2)}} (E \cdot B)_{J_2 \cdots J_p}{}^{J_2 \cdots J_{2n-1} L_{2n} \cdots L_{D-(p+2)}}$$
$$+ (-1)^{2n} c_{2n} (E \cdot B)^{L_2 \cdots L_{2n} J_{2n+1} \cdots J_p}{}_{L_2 \cdots L_{D-(p+2)}} (E \cdot B)_{J_2 \cdots J_p}{}^{J_2 \cdots J_{2n} L_{2n+1} \cdots L_{D-(p+2)}} , \tag{24}$$

where we denoted $c_r = C_r^p C_r^{D-(p+2)}$. The second line combines with the $J_{2n} = L_{2n}$ components of the first. In other words,

$$\sum_{r=1}^{r_{\max}} (-1)^r c_r (E \cdot B)^{L_2 \cdots L_r J_{r+1} \cdots J_p}{}_{L_2 \cdots L_{D-(p+2)}} (E \cdot B)_{J_2 \cdots J_p}{}^{J_2 \cdots J_r L_{r+1} \cdots L_{D-(p+2)}}$$
$$= \sum_{r \in 2\mathbb{Z}}^{r_{\max}} \tilde{c}_r (E \cdot B)^{L_2 \cdots L_r J_{r+1} \cdots J_p}{}_{L_2 \cdots L_{D-(p+2)}} (E \cdot B)_{J_2 \cdots J_p}{}^{J_2 \cdots J_r L_{r+1} \cdots L_{D-(p+2)}}$$
$$- \sum_{r \in 2\mathbb{Z}-1}^{r_{\max}} c_r \sum_{L_{r+1} \neq J_{r+1}} (E \cdot B)^{L_2 \cdots L_r J_{r+1} \cdots J_p}{}_{L_2 \cdots L_{D-(p+2)}} (E \cdot B)_{J_2 \cdots J_p}{}^{J_2 \cdots J_r L_{r+1} \cdots L_{D-(p+2)}} , \tag{25}$$

where

$$\tilde{c}_r = c_{r+1} - c_r = \left( \frac{p(D-(p+2)) - rD - 1}{(r+1)^2} \right) c_r . \tag{26}$$

Since $r \leq \min(p, D-(p+2))$, $p < D-2 < D$, and $D-(p+2) < D$, we have $\tilde{c}_r < 0$. Thus, the only positive contribution from the modulus of the Umov-Poynting vector to $V^2$ is the term proportional to $E^2 B^2$:

$$V^2 = (u^2)^2 \left[ \frac{u^2}{4} \left( p! E^2 + \frac{B^2}{(D-(p+2))!} \right)^2 - \frac{u^2 p!}{(D-(p+2))!} E^2 B^2 \right.$$
$$- \frac{u^2 p!}{(D-(p+2))!} \left( \sum_{r \in 2\mathbb{Z}}^{r_{\max}} \tilde{c}_r (E \cdot B)^{L_2 \cdots L_r J_{r+1} \cdots J_p}{}_{L_2 \cdots L_{D-(p+2)}} (E \cdot B)_{J_2 \cdots J_p}{}^{J_2 \cdots J_r L_{r+1} \cdots L_{D-(p+2)}} \right.$$
$$\left. \left. - \sum_{r \in 2\mathbb{Z}-1}^{r_{\max}} c_r \sum_{L_{r+1} \neq J_{r+1}} (E \cdot B)^{L_2 \cdots L_r J_{r+1} \cdots J_p}{}_{L_2 \cdots L_{D-(p+2)}} (E \cdot B)_{J_2 \cdots J_p}{}^{J_2 \cdots J_r L_{r+1} \cdots L_{D-(p+2)}} \right) \right] . \tag{27}$$

But the last term in the first line combines with the first term to form a perfect square, such that

$$V^2 = (u^2)^3 \left[ \frac{1}{4} \left( p! E^2 - \frac{B^2}{(D-(p+2))!} \right)^2 \right.$$
$$+ \frac{p!}{(D-(p+2))!} \left( \sum_{r \in 2\mathbb{Z}}^{r_{\max}} |\tilde{c}_r| (E \cdot B)^{L_2 \cdots L_r J_{r+1} \cdots J_p}{}_{L_2 \cdots L_{D-(p+2)}} (E \cdot B)_{J_2 \cdots J_p}{}^{J_2 \cdots J_r L_{r+1} \cdots L_{D-(p+2)}} \right.$$
$$\left. \left. + \sum_{r \in 2\mathbb{Z}-1}^{r_{\max}} c_r \sum_{L_{r+1} \neq J_{r+1}} (E \cdot B)^{L_2 \cdots L_r J_{r+1} \cdots J_p}{}_{L_2 \cdots L_{D-(p+2)}} (E \cdot B)_{J_2 \cdots J_p}{}^{J_2 \cdots J_r L_{r+1} \cdots L_{D-(p+2)}} \right) \right] . \tag{28}$$

The terms inside the square bracket above are manifestly positive definite. Since $u^M$ is causal, we conclude that $V^2 \leq 0$, and the DEC for $p$-form fields is satisfied for any background metric.

## 2.2 $p$-branes

In this section, we show that any membrane-like source, e.g. F-strings or (anti) D$p$-branes with action

$$S = -T_p \int d^{p+1}\xi \, e^{\alpha\phi} \sqrt{-h} \,, \qquad h_{ab} = g_{MN}\partial_a X^M \partial_b X^N \,, \tag{29}$$

where $X^M(\xi^a)$ are the embedding coordinates, satisfy the strong, null, weak, and dominant energy conditions if the tension $T_p$ is positive. The Wess-Zumino (WZ) coupling term is not necessary for our purposes, because it is topological and it does not contribute to the energy-momentum tensor.

To calculate the energy-momentum tensor of such sources, we need first a spacetime action:

$$S[X] = -T_p \int d^D x \int d^{p+1}\xi \, \delta^D(x - X(\xi)) \, e^{\alpha\phi} \sqrt{-h} \,. \tag{30}$$

Then,

$$T^{MN} = \frac{2}{\sqrt{-g}} \frac{\delta S}{\delta g_{MN}} = -\frac{T_p}{\sqrt{-g}} \int d^{p+1}\xi \, \delta^D(x - X(\xi)) \, e^{\alpha\phi} \sqrt{-h} \, h^{ab} \, \partial_a X^M \, \partial_b X^N \,. \tag{31}$$

To check for the validity of the energy conditions we need to study the sign of

$$h^{ab}\partial_a X^M \partial_b X^N u_M u_N \,,$$

where $u^M$ is timelike or null depending on the condition considered. We will again evoke the continuity of such contraction to cover the null case, and so from now on we assume $u^2 = g_{MN}u^M u^N < 0$. Since the energy-momentum tensor is localized, it suffices for our purposes to restrict $u^M$ to the worldvolume $\Sigma_{p+1}$ of the brane,

$$u^M(x(\tau))\big|_{\Sigma_{p+1}} = u^M(X(\xi)), \tag{32}$$

where $x^M(\tau)$ is the $\tau$-parametrized worldline whose tangent is $u^M$. In fact, when evaluating $T_{MN}$ on such a worldline, the delta-function will fix $x^M(\tau) = X^M(\xi^a)$, which can be solved to give $\xi = \xi(\tau)$. Another way of seeing this is to restrict the embedding map (suppressing the chart maps)

$$X: \quad \Sigma_{p+1} \to M_D$$
$$\xi^a \mapsto X^M(\xi^a),$$

to the worldline $x^M(\tau)$. In this case, the pullback of forms in $M_D$ will give

$$d\xi_a = \partial_a X^M dx_M \implies \dot{\xi}_a = \partial_a X^M u_M \,, \tag{33}$$

where we used a dot to denote derivative with respect to $\tau$. So, we have

$$h^{ab}\partial_a X^M \partial_b X^N u_M u_N \big|_{\Sigma_{p+1}} = h_{ab}\,\dot{\xi}^a \dot{\xi}^b \,. \tag{34}$$

On the other hand, restricting to the brane worldvolume and using the inverse Jacobian, which always exists for the relevant restricted codomain of $X$, we find that

$$u^2\big|_{\Sigma_{p+1}} = g_{MN}\partial_a X^M \partial_b X^N \dot{\xi}^a \dot{\xi}^b = h_{ab}\,\dot{\xi}^a \dot{\xi}^b \,. \tag{35}$$

Hence we conclude that

$$h^{ab}\partial_a X^M \partial_b X^N u_M u_N \big|_{\Sigma_{p+1}} = u^2\big|_{\Sigma_{p+1}} \,, \tag{36}$$

as it should be since $u^2$ restricted to the brane can also be computed by using the induced tangent vector and metric.

Therefore, if $T_p > 0$, we have

$$T_{MN} u^M u^N = -\frac{T_p}{\sqrt{-g}} \int d^{p+1}\xi\, \delta^D(x - X(\xi))\, e^{\alpha\phi}\, \sqrt{-h}\, u^2\big|_{\Sigma_{p+1}} > 0\,, \tag{37}$$

and both WEC and NEC are satisfied.

To test the SEC validity, we need to compute the energy-momentum tensor's trace

$$\begin{aligned}
T &= g_{MN} T^{MN} \\
&= -\frac{T_p}{\sqrt{-g}} \int d^{p+1}\xi\, \delta^D(x - X(\xi))\, e^{\alpha\phi}\, \sqrt{-h}\, h^{ab} \partial_a X^M \partial_b X^N g_{MN} \\
&= -(p+1)\frac{T_p}{\sqrt{-g}} \int d^{p+1}\xi\, \delta^D(x - X(\xi))\, e^{\alpha\phi}\, \sqrt{-h}\,, 
\end{aligned} \tag{38}$$

and so

$$\left(T_{MN} - \frac{T}{D-2} g_{MN}\right) u^M u^N = -\left(1 + \frac{p+1}{D-2}\right)\frac{T_p}{\sqrt{-g}} \int d^{p+1}\xi\, \delta^D(x - X(\xi)) e^{\alpha\phi} \sqrt{-h}\, u^2\big|_{\Sigma_{p+1}}\,, \tag{39}$$

which has the same sign as the brane's tension. So, for $T_p > 0$, the SEC is satisfied.

Finally, to show that DEC holds, we need to prove that the energy-flux vector

$$B^M = -T^M{}_N u^N\,, \tag{40}$$

is causal. Using the expression for $T^{MN}$ we get

$$B^M\big|_{\Sigma_{p+1}} = \frac{T_p}{\sqrt{-g}}\, e^{\alpha\phi}\, \sqrt{-h}\, h^{ab}\, \partial_a X^M\, \dot{\xi}_b\,, \tag{41}$$

where we have restricted $B^M$ to $\Sigma_{(p+1)}$, since $T_{MN}$ is localized there. So,

$$B_M B^M\big|_{\Sigma_{p+1}} = \left[T_p e^{\alpha\phi}\frac{\sqrt{-h}}{\sqrt{-g}}\right]^2 h_{ab}\dot{\xi}^a\dot{\xi}^b < 0\,, \tag{42}$$

*i.e.*, $B^M$ is timelike at events it does not vanish. We conclude that the DEC is also satisfied.

From the worldvolume perspective, the brane action defines a $(p+1)$-dimensional theory whose action describes a constrained system of worldvolume scalar fields. So, the spacetime results above are a manifestation of the fact that such a worldvolume theory cannot violate the discussed energy conditions. This viewpoint is useful for generalizing our conclusions when we turn on gauge fields and consider the pullback of spacetime fields on the worldvolume. As long as the worldvolume field configurations do not violate the energy conditions, we expect the spacetime conditions to be valid. Indeed, the action

$$S = -T_p \int d^{p+1}\xi\, e^{\alpha\phi}\sqrt{-\det(h + \mathcal{F})}\,, \quad \mathcal{F}_{ab} := F_{ab} + B_{ab}\,, \tag{43}$$

where $F_{ab}$ is the field strength for a worldvolume gauge field and $B_{ab}$ is the pullback of a spacetime 2-form field, has a $(p+1)$-dimensional energy-momentum tensor

$$\mathcal{T}^{ab} = -T_p\, e^{\alpha\phi}\, \frac{\sqrt{-\mathcal{G}}}{\sqrt{-h}}\, (\mathcal{G}^{-1})^{ab}\,, \tag{44}$$

where $\mathcal{G}_{ab} := h_{ab} + \mathcal{F}_{ab}$. Thus, the spacetime energy-momentum tensor coming from the brane and worldvolume fields configurations can be written as

$$
\begin{aligned}
T^{MN} &= -\frac{T_p}{\sqrt{-g}} \int d^{p+1}\xi \, \delta^D(x - X(\xi)) \, e^{\alpha\phi} \, \sqrt{-\mathcal{G}} \left(\mathcal{G}^{-1}\right)^{ab} \partial_a X^M \partial_b X^N \\
&= \frac{1}{\sqrt{-g}} \int d^{p+1}\xi \, \delta^D(x - X(\xi)) \, \sqrt{-h} \, \mathcal{T}^{ab} \, \partial_a X^M \partial_b X^N \,.
\end{aligned}
\tag{45}
$$

Hence, the validity of the spacetime energy conditions follows from the validity of the worldvolume energy conditions. In particular, $\mathcal{T}_{ab}\dot{\xi}^a\dot{\xi}^b$ and $\mathcal{T}_{ab}h^{ab}$ will determine the signs of $T_{MN}u^m u^N$ and $T_{MN}g^{MN}$, although $\mathcal{T}^{ab}$ is not the pullback of $T^{MN}$.

## 2.3 O$p$-planes

From the results of the previous section, it is clear that objects with negative tension, $T_p < 0$, violate the energy conditions previously discussed. Dynamical-extended objects with negative tension would give rise to instabilities manifested as tachyons in the worldvolume theory. However, string compactification on orientifolds produces extra source terms in the action for each fixed locus of the orientifold group. Although non-dynamical, these O$p$-planes couple to the $p$-forms, metric, and dilaton fields, and so they contribute to the energy-momentum tensor of the theory. It turns out that orientifold planes have negative tension without introducing instabilities because they do not carry extra dynamical degrees of freedom. For that reason, O$p$-planes are not really extended physical objects, but rather a manifestation of the background structure of orientifolds. Fields defined over such backgrounds need to satisfy some consistency conditions, and these are taken into account by their coupling to the O$p$-planes,

$$
S = -\sum_i T_{\mathrm{O}p}^i \int d^D x \int d^{p+1}\xi \, \delta^D(x - X_i(\xi)) \, e^{\beta\phi} \, \sqrt{-h_i} + \sum_i \mu_{\mathrm{O}p}^i \int A_{p+1} \wedge *J_{p+1}^i \,,
\tag{46}
$$

where we used the index $i$ to denote individual fixed planes and included the WZ coupling to a $(p+1)$-form field $A_{p+1}$. The current $*J_{p+1}^i$ is localized at the $i$-th fixed plane. For $T_{\mathrm{O}p}^i < 0$, the energy conditions of the previous section are violated.

However, if we have both D$p$-branes and O$p$-planes in the theory, we might hope that the positive tension of the former counterbalances the negative tension of the latter, in such a way that the energy conditions might still hold true, that is, the sign of the contractions needed to prove the energy conditions depend on $\sum_a T_p^a$ and it could be that this total sum over orientifold-planes and D$p$-branes is positive. This mechanism will depend on the number and orientation of the sources.

Indeed, in string theory, the number of D$p$-branes and O$p$-planes are constrained by the tadpole cancellation condition condition as can be seen from the WZ coupling. Consistency with the $(p+1)$-form equation of motion implies that the source term for $A_{p+1}$ should be closed, and as a consequence

$$
\sum_a \mu_p^a \int_{C_{D-p-1}} *J_{p+1}^a = 0 \,,
\tag{47}
$$

for any submanifold without boundary $C_{D-p-1}$ transverse to the worldvolume of the sources. Note that we have also included the D$p$-branes contribution into the sum. Considering the case of parallel sources, we get the charge cancellation conditions

$$
\sum_a \mu_p^a = \sum_i \left(\mu_{\mathrm{D}p}^i + \mu_{\mathrm{O}p}^i\right) = 0 \,.
\tag{48}
$$

We denote $Op_\pm$ orientifold planes with the same (opposite) orientation compared to $Dp$-branes, so effectively $Op_-$-planes have positive tension, *i.e.* $T_{Op_\pm} = \mp|\mu_{Op_\pm}|$. Generically, the charge of a $Op_+$ or $Op_-$-plane is related to the $Dp$-brane charge as $\mu_{Op_\pm} = \mp 2^{p-5}\mu_{Dp}$, and anti-orientifold planes have the opposite charge. Since $Dp$-branes are extremal objects, we have $T_{Op_\pm} = \mp 2^{p-5} T_{Dp}$.

Let us write the tadpole cancellation condition explicitly for a system having $N_{Dp}$ $Dp$-branes, $N_{Op_+}$ $Op_+$-planes, $N_{Op_-}$ $Op_-$-planes, $N_{\overline{Dp}}$ $\overline{Dp}$-(anti)branes, $N_{\overline{Op_+}}$ $\overline{Op}_+$-(anti)planes and $N_{\overline{Op_-}}$ $\overline{Op}_-$-(anti)planes. Then we have

$$\sum_a \mu_p^a = \mu_{Dp}\left[ N_{Dp} - N_{\overline{Dp}} - 2^{p-5}\left( N_{Op_+} + N_{\overline{Op_-}} - N_{\overline{Op_+}} - N_{Op_-} \right)\right] = 0, \tag{49}$$

which should be valid for each value of $p$ (given a supergravity action, for some $p$ it will be automatically satisfied). All physical configurations containing parallel $Dp$-branes and $Op$-planes should satisfy the condition above. But due to the relation between tension and charge of the various sources, the charge cancellation condition implies

$$\sum_a T_p^a = T_{Dp}\left[ N_{Dp} + N_{\overline{Dp}} - 2^{p-5}\left( N_{Op_+} - N_{\overline{Op_-}} + N_{\overline{Op_+}} - N_{Op_-} \right)\right]$$
$$= 2T_{Dp}\left[ N_{Dp} - 2^{p-5}(N_{Op_+} - N_{Op_-}) \right]. \tag{50}$$

So, the requirement for having a non-negative total tension might be written as

$$\sum_a T_p^a \geq 0 \iff N_{Dp} \geq 2^{p-5}(N_{Op_+} - N_{Op_-}). \tag{51}$$

For supersymmetry preserving solutions, all the $Op$-planes have the same orientation as the $Dp$-branes ($N_{Op_-} = 0$) and $N_{Dp} = 2^{p-5}N_{Op_+}$. In those cases, the total tension vanishes and the higher dimensional weak, strong, null, and dominant energy conditions are satisfied. In summary, although orientifold planes might violate the energy conditions, the consistency of the equations of motion stringently constrains the physical configurations to satisfy the energy conditions.

## 3 Energy conditions inheritance

In this section, we point out sufficient conditions for a $D$-dimensional energy condition to imply a lower $d$-dimensional one. We call this relation the *inheritance of energy conditions*. In particular, we are interested in energy conditions with respect to the $d$-dimensional metric $\tilde{g}_{\mu\nu}(x)$, which appears in the $D$-dimensional one as

$$d\bar{s}^2 = \overline{g}_{MN}dx^M dx^N$$
$$= \Omega^2(y)\tilde{g}_{\alpha\beta}(x)dx^\alpha dx^\beta + h_{mn}(x,y)dy^m dy^n, \tag{52}$$

where the internal volume does not dependent on $x^\mu$. We use the results presented in appendix A to write the $D$-dimensional quantities in terms of the $d$-dimensional ones.

### 3.1 Null energy condition

The $D$-dimensional geometric NEC is given by

$$\overline{R}_{MN}l^M l^N \geq 0, \qquad \overline{g}_{MN}l^M l^N = 0. \tag{53}$$

In the following, we show that assuming an internal space with constant volume is sufficient for the $D$-dimensional NEC to imply the $d$-dimensional NEC.

Specifying the null vector $l^M$ to have components only in the lower $d$-dimensional tangent bundle, we have

$$\overline{R}_{\mu\nu}l^\mu l^\nu \geq 0 \,. \tag{54}$$

Using the expression (A.3a) for $\overline{R}_{\mu\nu}$, this implies that

$$\tilde{R}_{\alpha\beta}(\tilde{g})l^\alpha l^\beta + \frac{1}{4}\tilde{\nabla}_\alpha h^{pq}\tilde{\nabla}_\beta h_{pq}l^\alpha l^\beta \geq 0 \,. \tag{55}$$

Employing the identity

$$\tilde{\nabla}_\alpha h^{pq} = -h^{pm}h^{qn}\tilde{\nabla}_\alpha h_{mn} \,, \tag{56}$$

we can rewrite the second term on the left-hand side as

$$-h^{pm}h^{qn}\tilde{\nabla}_\alpha h_{mn}\tilde{\nabla}_\beta h_{pq}l^\alpha l^\beta = -h^{mp}(l^\alpha\tilde{\nabla}_\alpha h_{mn})h^{nq}(l^\beta\tilde{\nabla}_\beta h_{pq}) = -\mathrm{tr}\left[h^{-1}(l\cdot\tilde{\nabla}h)\right]^2 \,. \tag{57}$$

As long as $h_{mn}$ is positive-definite and $h^{mn}$ and $\tilde{\nabla}h_{mn}$ are well-defined, this term is negative semi-definite. Hence, we get

$$\tilde{R}_{\alpha\beta}(\tilde{g})l^\alpha l^\beta \geq +\frac{1}{4}h^{mp}(l^\alpha\tilde{\nabla}_\alpha h_{mn})h^{nq}(l^\beta\tilde{\nabla}_\beta h_{pq}) \geq 0 \,, \tag{58}$$

which is the $d$-dimensional NEC for the metric $\tilde{g}$.

## 3.2  Strong energy condition

The $D$-dimensional geometric SEC is given by

$$\overline{R}_{MN}u^M u^N \geq 0 \,, \qquad \overline{g}_{MN}u^M u^N < 0 \,. \tag{59}$$

Restricting $u^M$ to the $d$-dimensional tangent bundle, we have

$$\tilde{R}_{\alpha\beta}(\tilde{g})u^\alpha u^\beta + \frac{1}{4}\tilde{\nabla}_\alpha h^{pq}\tilde{\nabla}_\beta h_{pq}u^\alpha u^\beta - u^2\frac{\Omega^{-(d-2)}}{d}\nabla^2\Omega^d \geq 0 \,, \tag{60}$$

or

$$\tilde{R}_{\alpha\beta}(\tilde{g})u^\alpha u^\beta \geq -\frac{1}{4}\tilde{\nabla}_\alpha h^{pq}\tilde{\nabla}_\beta h_{pq}u^\alpha u^\beta + u^2\frac{\Omega^{-(d-2)}}{d}\nabla^2\Omega^d \,. \tag{61}$$

The second term is positive-definite for the same reason as in the NEC case:

$$\tilde{R}_{\alpha\beta}(\tilde{g})u^\alpha u^\beta \geq \frac{1}{4}h^{mp}(u^\alpha\tilde{\nabla}_\alpha h_{mn})h^{nq}(u^\beta\tilde{\nabla}_\beta h_{pq}) + u^2\frac{\Omega^{-(d-2)}}{d}\nabla^2\Omega^d \geq u^2\frac{\Omega^{-(d-2)}}{d}\nabla^2\Omega^d \,. \tag{62}$$

Multiplying the resulting expression by $\Omega^{(d-2)}$ and integrating over the internal manifold we get

$$\frac{G_D}{G_d}\tilde{R}_{\alpha\beta}(\tilde{g})u^\alpha u^\beta \geq 0 \,, \tag{63}$$

where we have used the fact that the internal manifold is compact and the relation between the higher and lower-dimensional Newton's constants

$$\frac{G_D}{G_d} = \int d^{(D-d)}y\,\sqrt{h}\,\Omega^{(d-2)} > 0 \,. \tag{64}$$

Hence, we have shown that

$$\tilde{R}_{\alpha\beta}(\tilde{g})u^\alpha u^\beta \geq 0 \,, \qquad \tilde{g}_{\alpha\beta}u^\alpha u^\beta < 0 \,, \tag{65}$$

which is the $d$-dimensional SEC.

## 3.3 Weak energy condition

The $D$-dimensional WEC states that

$$\overline{G}_{MN}u^M u^N \geq 0\,, \qquad \overline{g}_{MN}u^M u^N < 0\,. \tag{66}$$

Restricting $u^M$ to the $d$-dimensional tangent bundle and using (A.4a), we get

$$\overline{G}_{\alpha\beta}u^\alpha u^\beta = \tilde{G}_{\alpha\beta}u^\alpha u^\beta - \frac{1}{2}u^2\Omega^2 R(h) + \left(\frac{1}{4}\tilde{\nabla}_\alpha h^{pq}\tilde{\nabla}_\beta h_{pq} - \frac{1}{8}\tilde{g}_{\alpha\beta}\tilde{\nabla}^\sigma h^{pq}\tilde{\nabla}_\sigma h_{pq}\right)u^\alpha u^\beta$$
$$+ \frac{2(d-1)}{d}u^2\Omega^{2-d/2}\nabla^2\Omega^{d/2} \geq 0\,. \tag{67}$$

Defining

$$T_{\alpha\beta}^{(h)} := -\frac{1}{4}\tilde{\nabla}_\alpha h^{pq}\tilde{\nabla}_\beta h_{pq} + \frac{1}{8}\tilde{g}_{\alpha\beta}\tilde{\nabla}^\sigma h^{pq}\tilde{\nabla}_\sigma h_{pq}\,, \tag{68a}$$

$$T_{\alpha\beta}^{(\Omega)} := \left(\frac{1}{2}\Omega^2 R(h) - \frac{2(d-1)}{d}\Omega^{2-d/2}\nabla^2\Omega^{d/2}\right)\tilde{g}_{\alpha\beta} =: \Lambda(y)\tilde{g}_{\alpha\beta}\,, \tag{68b}$$

we can rewrite (67) as

$$\tilde{G}_{\alpha\beta}u^\alpha u^\beta \geq T_{\alpha\beta}^{(h)}u^\alpha u^\beta + T_{\alpha\beta}^{(\Omega)}u^\alpha u^\beta\,. \tag{69}$$

It will become clear soon that it makes sense to interpret $T_{\alpha\beta}^{(h)}$ and $T_{\alpha\beta}^{(\Omega)}$ as energy-momentum tensors. Hence, the $D$-dimensional weak energy condition implies the lower dimensional one provided both $T_{\alpha\beta}^{(h)} + T_{\alpha\beta}^{(\Omega)}$ satisfy the $d$-dimensional WEC. But since

$$T_{\alpha\beta}^{(\Omega)}u^\alpha u^\beta \geq 0 \implies \Lambda(y) = \left(\frac{1}{2}\Omega^2 R(h) - \frac{2(d-1)}{d}\Omega^{2-d/2}\nabla^2\Omega^{d/2}\right) \leq 0\,, \tag{70}$$

after multiplying by $\Omega^{d/2-2}$ and integrating over the compact internal manifold, the WEC for $T_{\alpha\beta}^{(\Omega)}$ implies

$$\int d^{D-d}y\,\sqrt{h}\,\Omega^{d/2}R(h) \leq 0\,, \tag{71}$$

which is trivially satisfied for Ricci-flat internal spaces. The same trick could be used in (69) already, and so the WEC for $T_{\alpha\beta}^{(h)}$ and an internal manifold with negative mean Ricci curvature (weighted by $\Omega^{d/2}$) are sufficient conditions for WEC inheritance.

Note that $T_{\alpha\beta}^{(h)}$ is the energy-momentum tensor for the moduli fields $h_{mn}$, which can be seen from the dimensional reduction for the $D$-dimensional Einstein-Hilbert action:

$$\int d^D x\,\sqrt{-\overline{g}}\,\overline{R}^{(D)} \supset \int d^d x\,\tilde{V}_6\,\sqrt{-\tilde{g}}\left(R(\tilde{g}) + \frac{1}{4}\nabla^\alpha h^{pq}\nabla_\alpha h_{pq} + \frac{1}{4}h^{pq}h_{ns}\nabla^\alpha h_{pq}\nabla_\alpha h^{sn} + \cdots\right)\,, \tag{72}$$

where we assumed an internal manifold with constant total volume. In particular, if all $h_{mn}$ are constant, the WEC for $T_{\alpha\beta}^{(h)}$ is automatically satisfied.

## 3.4 Dominant energy condition

The $D$-dimensional DEC states that

$$\overline{G}_{MN}u^M u^N \geq 0\,, \qquad \overline{g}_{MN}u^M u^N < 0\,, \qquad \overline{g}_{MN}\overline{G}^M_{\ P}\overline{G}^N_{\ Q}u^P u^Q \leq 0\,, \tag{73}$$

which are the WEC and the condition for $-G^M_{\ N}u^N$ to be a causal vector (timelike or null). We already know the conditions under which $\tilde{G}_{\alpha\beta}$ obeys the lower-dimensional WEC, so we will

focus on the flux condition for $\tilde{G}^\alpha{}_\beta u^\beta$, which has to be timelike or lightlike. Assuming $u^P$ to be on the $d$-dimensional tangent bundle as before, we get

$$\overline{g}_{MN}\overline{G}^M{}_P\overline{G}^N{}_Q u^P u^Q = \left(\Omega^2 \tilde{g}_{\mu\nu}\overline{G}^\mu{}_\alpha\overline{G}^\nu{}_\beta + h_{mn}\overline{G}^m{}_\alpha\overline{G}^n{}_\beta\right)u^\alpha u^\beta\,, \tag{74}$$

and so

$$\overline{g}_{MN}\overline{G}^M{}_P\overline{G}^N{}_Q u^P u^Q \leq 0 \implies \Omega^2 \tilde{g}_{\mu\nu}\overline{G}^\mu{}_\alpha\overline{G}^\nu{}_\beta u^\alpha u^\beta \leq -h_{mn}\overline{G}^m{}_\alpha\overline{G}^n{}_\beta u^\alpha u^\beta \leq 0\,, \tag{75}$$

where we have used the fact that $h_{mn}$ is positive definite. Note that we can write the $d$-dimensional components of $\overline{G}_{MN}$ as

$$\overline{G}_{\rho\alpha}u^\alpha = V_\rho - U_\rho - W_\rho\,, \tag{76}$$

where we have defined

$$V_\rho := \tilde{G}_{\rho\alpha}u^\alpha\,, \qquad U_\rho := T^{(h)}_{\rho\beta}u^\beta\,, \qquad W_\rho := T^{(\Omega)}_{\rho\beta}u^\beta\,. \tag{77}$$

For the lower-dimensional WEC to be satisfied, we saw in the last section that $T^{(h)}_{\alpha\beta} + T^{(\Omega)}_{\alpha\beta}$ must satisfy the $d$-dimensional WEC. But due to the form of $T^{(\Omega)}_{\alpha\beta}$ we have

$$T^{(\Omega)}_{\rho\alpha}T^{(\Omega)\rho}{}_\beta u^\alpha u^\beta = \Lambda^2(y)\tilde{g}_{\rho\alpha}\delta^\rho_\beta u^\alpha u^\beta = \Lambda^2(y)u^2 \leq 0\,, \tag{78}$$

because $u^2 < 0$, which means that the vector $W^\rho = \tilde{g}^{\rho\sigma}W_\sigma$ is necessarily timelike or lightlike. Assuming also that $T^{(h)}_{\alpha\beta}$ satisfy the DEC, we have that $U^\rho = \tilde{g}^{\rho\sigma}U_\sigma$ is also timelike or lightlike. Now, the higher-dimensional DEC can be rewritten as (see (75))

$$\tilde{g}_{\mu\nu}(V^\mu - U^\mu - W^\mu)(V^\nu - U^\nu - W^\nu) \leq 0\,, \tag{79}$$

where $V^\mu = \tilde{g}^{\mu\sigma}V_\sigma$. We now show that this last equality, together with the properties of $U$ and $W$, implies that $V$ is timelike or lightlike.

Firstly, let us restructure $V^\mu$ as $V^\mu = V^\mu + T^\mu - T^\mu$, where $T^\mu = U^\mu + W^\mu$. Then, we have

$$V^2 = \tilde{g}_{\mu\nu}V^\mu V^\nu = (V-T)^2 + T^2 + 2T_\mu(V^\mu - T^\mu) \leq T^2 + 2T_\mu(V^\mu - T^\mu)\,, \tag{80}$$

where we have used the higher-dimensional DEC. Secondly, let's use the fact that if two vectors $A$ and $B$ are timelike (or lightlike) and both future or past directed, then $A^\mu B_\mu \leq 0$ and $A+B$ is also timelike (or lightlike),[2] i.e., if $A$ and $B$ are in the same local causal cone to $p$, then $A+B$ is also in the causal cone. Applying this to $T = U + W$ we have $T^2 \leq 0$ and $T$ is also in the same causal cone as $U$ and $W$ and so is $V-T$ as a consequence of the higher-dimensional DEC (79). Then,

$$V^2 \leq 2T_\mu(V^\mu - T^\mu) \leq 0 \iff \tilde{g}_{\mu\nu}\tilde{G}^\mu{}_\alpha\tilde{G}^\nu{}_\beta u^\alpha u^\beta \leq 0\,, \tag{81}$$

i.e., the lower-dimensional DEC. The equality holds when $U$, $V$, and $W$ are all proportional and lightlike (or when they all vanish).

---

[2]For our purposes, we consider that the vectors $A, B \in T_p M$ are tangent to geodesics connecting $p$ to its causal future $J^+(p)$ or past $J^-(p)$, i.e. they are cooriented. Then, $A$ and $B$ are in the future (past) local causal cone on $p$ if and only if $A^\mu B_\mu \leq 0$, essentially because $A^\mu B_\mu$ and $A^0 B_0$ have opposite signs, with the equality holding for $A \propto B$ and both lightlike (unless one of them is zero) [36]. Then, the fact that $A+B$ is also on the same causal cone follows from $(A+B)^2 = A^2 + B^2 + 2A^\mu B_\mu \leq 2A^\mu B_\mu \leq 0$ and $A^\mu(A_\mu + B_\mu) \leq 0$.

# 4  Discussion and Conclusion

In trying to obtain cosmological backgrounds from string theory, it is important to identify the allowed solutions from different compactifications. Although there has been a lot of work done in this direction in the past, it has mainly been to find dS space in four dimensions. This is certainly understandable, given the current stage of accelerated expansion of our universe. Nevertheless, we believe it is worthwhile to take a more holistic view while trying to test the viability of cosmological compactifications. One way to do so is to focus on the various energy conditions since these have the potential to systematically classify different backgrounds. Another way of saying this is to focus on the full covariant energy-momentum tensor and not just on its trace or specific components as might be relevant for particular solutions (such as dS).

Given this motivation, our findings have been two-fold. Firstly, we showed that the low-energy limit of string theory, described by various supergravity actions, consists of an action that does not violate the various energy conditions.[3] And secondly, we showed how these higher-dimensional energy conditions typically imply the corresponding lower-dimensional ones. Quite naturally, both these findings come with their own set of assumptions.

For the validity of the energy conditions in supergravity theories, we assumed that:

1. No classical higher derivative (or $\alpha'$) curvature corrections were taken into account. In fact, it is known that some of the energy conditions might indeed be violated when considering non-perturbative solutions in the presence of a tower of $\alpha'$-corrections [42, 43] (see also [44]).

2. A dilaton potential is absent in the higher-dimensional supergravity action and we are not considering the massive type IIA theory or actions from non-supersymmetric string theories (no-go theorems for the latter were studied in [19]).

3. We are also ignoring all types of (perturbative and non-perturbative) quantum corrections which might arise from string theory. Once again, it has been shown that quantum corrections can indeed give rise to cosmological solutions which violate some of these energy conditions, when considering an M-theory uplift of type IIB string theory [10].

Given these assumptions, what we have been able to show is that the various terms present in the supergravity action do satisfy the energy conditions. In fact, a surprising result was that even when some of these individual ingredients *do* violate the energy conditions, consistency conditions require that the overall physical configuration obeys them. In particular, we found that O$p$-planes, due to their negative tension, end up violating the energy conditions on their own, along expected lines. However, the tadpole cancellation condition requires the presence of D$p$-branes such that overall the energy conditions are *still* obeyed. A systematic study of the condition for non-negative total tension for more general D-brane configurations on orientifolds (along the lines of [45], for instance) is left for future work. Although unrelated, note that the calculations in section 2.3 do not include flux contributions to the tadpole cancellation. They would come from couplings to other fluxes in the WZ term[4] and would change the left-hand side (51) by an extra term proportional to $N_{\text{flux}}/\mu_{\text{D}p}$.

---

[3]We have assumed the standard field content from the massless string spectrum and usual sources like D-branes. Other states, such as S-branes [37] (see [38–41] for cosmological applications), might violate some of the energy conditions.

[4]Due to their topological property, such an extra coupling will not affect the results of sections 2.1 and 2.2 on the validity of energy conditions for p-forms and extended objects since we have not assumed an explicit form of the $p$-form's Bianchi identity. The conclusions in section 2.3 will be slightly modified because of extra terms in equation (48) that will induce extra terms in (51): the tadpole cancellation has to be considered for the validity of the energy conditions, but the requirement for positive total tension (51) that we found as a consequence of it is modified if fluxes are included. Discussing these modifications further is beyond the scope of this work.

The main result of the present work might be summarized by the following assertion:

*Energy conditions inheritance: If a given energy condition is satisfied in a higher D-dimensional manifold, then, given some reasonable assumptions, the corresponding lower d-dimensional energy condition is also respected.*

This follows after assuming the *D*-dimensional metric to have a decomposition of the form (52). We emphasize that this is a rather mild assumption, as this is the type of metric that is assumed in all standard flux compactifications (such as [46, 47]), but also more general since it allows for a *x*-dependent internal metric. The other reasonable assumptions depend on which energy condition is considered. We have seen that:

- A constant internal volume and positive well-defined internal metric are sufficient conditions for the NEC and the SEC inheritance, *i.e.*, assuming Newton's constant to remain constant and a non-singular internal space are sufficient conditions for these two energy conditions to be respected in the *d*-dimensional theory.

- The tensor $T_{\alpha\beta} = T_{\alpha\beta}^{(h)} + T_{\alpha\beta}^{(\Omega)}$ (defined in (68)) has to satisfy WEC for us to have WEC inheritance. The simplest possibility is for $T^{(h)}$ and $T^{(\Omega)}$ to satisfy WEC independently. The $T_{\alpha\beta}^{(h)}$ has an interpretation in terms of moduli fields $h_{mn}$, and thus we find that the WEC has to be valid for these fields. On the other hand, $T_{\alpha\beta}^{(\Omega)}$ has an interpretation in terms of the geometry of the internal manifold, and it satisfying the WEC implies condition (71) on the curvature of the internal manifold.

- Since $T_{\alpha\beta}^{(\Omega)}u^\beta$ is necessarily causal, sufficient requirements for DEC inheritance are the WEC for $T_{\alpha\beta}^{(h)} + T_{\alpha\beta}^{(\Omega)}$ and that $T_{\alpha\beta}^{(h)}u^\beta$ causal, where $u^\beta$ is an arbitrary but timelike vector field.

Given our assumptions, our results can be interpreted as covariant consistency conditions that must be satisfied in flux compactifications. Since it might be interesting to study cosmological backgrounds which violate some of these energy conditions, that would imply going around some of the previous requirements. We hope that clearly stating our assumptions and the conclusions following them will help in the construction of low-energy backgrounds that violate a given energy condition.

## Acknowledgments

We thank Keshav Dasgupta for discussions and comments on a previous draft version and Radu Tatar for discussions and early involvement in this work. We also thank Takanao Tsuyuki for pointing out a mistake in a previous version of this work. HB would like to thank Jéssica Martins for discussions and the ETH-Zurich and the Higgs Centre for Theoretical Physics for their hospitality during the late-stage execution of this work.

For the purpose of open access, the authors have applied a Creative Commons Attribution (CC BY) license to any Author Accepted Manuscript version arising from this submission.

**Funding information**  HB's research is supported by the Fonds de Recherche du Québec (PBEEE/303549) and partially by funds from NSERC. SB is supported in part by the Higgs Fellowship and by the STFC Consolidated Grant "Particle Physics at the Higgs Centre". MMF's research is supported by NSERC and in part by the Delta Institute of Theoretical Physics.

## A Ricci and Einstein tensors

In this appendix, we recollect components of the Ricci and Einstein tensors relevant for the calculations on previous sections. We consider the $D$-dimensional metric $\bar{g}_{MN}$ to have the form

$$d\bar{s}^2 = \bar{g}_{MN}(x,y)dx^M dx^N = \Omega^2(y)\tilde{g}_{\mu\nu}(x)dx^\mu dx^\nu + h_{mn}(x,y)dy^m dy^n, \qquad (A.1)$$

and are after writing the components of the curvature tensors in terms of the ones corresponding to $\tilde{g}_{\mu\nu}$ and $h_{mn}$. Imposing the internal volume to be $x$-independent,

$$\nabla_\mu \sqrt{\det h} = 0 \implies h^{mn}\nabla_\mu h_{mn} = 0, \qquad \nabla_M h^{mn}\nabla_\mu h_{mn} = -h^{mn}\nabla_M \nabla_\mu h_{mn}, \qquad (A.2)$$

we have

$$\overline{R}_{\alpha\beta} = \tilde{R}_{\alpha\beta}(\tilde{g}) + \frac{1}{4}\tilde{\nabla}_\alpha h^{pq}\tilde{\nabla}_\beta h_{pq} - \frac{1}{d}\tilde{g}_{\alpha\beta}\Omega^{2-d}\nabla^2\Omega^d, \qquad (A.3a)$$

$$\overline{R}_{pq} = R_{pq}(h) - \frac{1}{2}\Omega^{-2}\tilde{g}^{\mu\rho}\tilde{\nabla}_\mu\tilde{\nabla}_\rho h_{pq} + \frac{1}{2}\Omega^{-2}h^{nr}\tilde{\nabla}_\rho h_{qr}\tilde{\nabla}^\rho h_{pn} - d\Omega^{-2}\nabla_p\nabla_q\Omega, \qquad (A.3b)$$

$$\overline{R}_{p\beta} = -\frac{1}{2}\nabla_p h_{ms}\tilde{\nabla}_\beta h^{sm} - \frac{1}{2}h_{mp}\nabla_s\tilde{\nabla}_\beta h^{sm} + \left(\frac{d}{2}-1\right)\Omega^{-1}\nabla^s\Omega\tilde{\nabla}_\beta h_{ps}, \qquad (A.3c)$$

$$\overline{R} = \Omega^{-2}\tilde{R}(\tilde{g}) + R(h) + \frac{1}{4}\Omega^{-2}\tilde{\nabla}^\alpha h^{pq}\tilde{\nabla}_\alpha h_{pq} - \frac{4d}{d+1}\Omega^{-\frac{d+1}{2}}\nabla^2\Omega^{\frac{d+1}{2}}, \qquad (A.3d)$$

and

$$\overline{G}_{\alpha\beta} = \tilde{R}_{\alpha\beta}(\tilde{g}) - \frac{1}{2}\tilde{g}_{\alpha\beta}\tilde{R}(\tilde{g}) - \frac{1}{2}\tilde{g}_{\alpha\beta}\Omega^2 R(h) + \frac{1}{4}\tilde{\nabla}_\alpha h^{pq}\tilde{\nabla}_\beta h_{pq} - \frac{1}{8}\tilde{g}_{\alpha\beta}\tilde{\nabla}^\sigma h^{pq}\tilde{\nabla}_\sigma h_{pq}$$
$$+ \frac{2(d-1)}{d}\tilde{g}_{\alpha\beta}\Omega^{2-\frac{d}{2}}\nabla^2\Omega^{\frac{d}{2}}, \qquad (A.4a)$$

$$\overline{G}_{pq} = R_{pq}(h) - \frac{1}{2}h_{pq}R(h) - \frac{1}{2}h_{pq}\Omega^{-2}\tilde{R}(\tilde{g}) - \frac{1}{2}\Omega^{-2}\tilde{\nabla}_\rho\tilde{\nabla}^\rho h_{pq} + \frac{1}{2}h^{nr}\Omega^{-2}\tilde{\nabla}_\rho h_{qr}\tilde{\nabla}^\rho h_{pn}$$
$$- \frac{1}{8}h_{pq}\Omega^{-2}\tilde{\nabla}^\alpha h^{mn}\tilde{\nabla}_\alpha h_{mn} - d\Omega^{-1}\nabla_p\nabla_q\Omega + \frac{2d}{d+1}h_{pq}\Omega^{-\frac{d+1}{2}}\nabla^2\Omega^{\frac{d+1}{2}}, \qquad (A.4b)$$

$$\bar{G}_{p\beta} = -\frac{1}{2}\nabla_p h_{ms}\tilde{\nabla}_\beta h^{sm} - \frac{1}{2}h_{mp}\nabla_s\tilde{\nabla}_\beta h^{sm} + \left(\frac{d}{2}-1\right)\tilde{\nabla}_\beta h_{ps}\Omega^{-1}\nabla^s\Omega. \qquad (A.4c)$$

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
