# Peer review of "The inheritance of energy conditions: Revisiting no-go theorems in string compactifications"

_SciPost Physics, doi:SciPost Phys. 15, 225 (2023)_

## Round 4 · Referee Report · Anonymous · 2023-8-21

Report

The authors have addressed all points in a satisfactory way, except the first one. In their derivation of the energy conditions for a configuration of Op planes and Dp branes, they use explicitly (Eqs 38 to 42) the tadpole cancelation condition without fluxes. Furthermore, since fluxes result in positive charges, the inequality in the right hand side of Equation (42) is precisely the opposite, the number of Dp branes has to be less than $2^{p-5}
\times N_{O_{p+}}$ (assuming there are no $O_{p-}$ planes). The authors should still clarify how the conclusions of this section are valid without using (38)-(41).

  • validity: -
  • significance: -
  • originality: -
  • clarity: -
  • formatting: -
  • grammar: -

Author:  Heliudson Bernardo  on 2023-09-07  [id 3960]

(in reply to Report 1 on 2023-08-21)

Dear Anonymous Referee,

Thank you again for reading the paper and for your constructive comments toward increasing the paper's applicability scope. We reply to your comment (copied below) about flux contribution as follows:

*The authors have addressed all points in a satisfactory way, except the first one. In their derivation of the energy conditions for a configuration of Op planes and Dp branes, they use explicitly (Eqs 38 to 42) the tadpole cancelation condition without fluxes. Furthermore, since fluxes result in positive charges, the inequality in the right hand side of Equation (42) is precisely the opposite, the number of Dp branes has to be less than 2p−5×NOp+ (assuming there are no Op− planes). The authors should still clarify how the conclusions of this section are valid without using (38)-(41).*

In section 2.2, we have shown that whether an extended object will satisfy the energy conditions or not depends on the sign of its tension. For D-branes, the tension is positive, and the conditions are satisfied. For a single $Op_+$-planes, the tension is negative, and the energy conditions are not satisfied. But if one considers a configuration containing sets of parallel D-branes and O-planes it might happen that the total tension is positive and the energy conditions are satisfied. However, the configuration of extended objects is constrained by the tadpole cancellation condition, which is a constraint on the charge of the system. Since in string theory the charge and tension of extended objects are related, this begs the question of how the tadpole cancellation condition affects the expression of the total tension of the configuration, and in section 2.3 we investigated precisely that.

If we consider a parallel configuration containing D-branes, anti-D-branes, O-planes, and anti-O-planes (including different orientations), the tadpole cancellation condition as coming from the minimal WZ coupling with the corresponding p-form (see last term in equation (37)) reduces to equation (40) and constrains the number of extended objects of each kind. The total tension of the system is given by the first line of equation (41). Upon using (40), the total tension reduces to the second line of equation (41). Thus, we conclude that, if we want the total tension to be positive for the energy conditions to hold, then we need the inequality in (42) to be satisfied.

Now let's discuss which of our results would change if we consider the "non-minimal", general coupling of extended objects to the p-forms
\begin{equation}
S_{\text{WZ}} = \mu_p \int_{\Sigma_{p+1}} e^{B_2 + 2\pi \alpha' F_2}\wedge \sum_i A_i.
\end{equation}
The exponential is a formal expression to collect terms with the required rank to match with $A_i$ and the dimension of the extended source. Note that the minimal coupling considered in the paper corresponds to the first term after expanding the exponential. The action above *does not* depend on the metric and, as such, it *will not* contribute to the energy-momentum tensors of the p-forms and of the extended objects. Such energy-momentum tensors were explicitly used in the discussions of sections 2.1 and 2.2. Therefore, we conclude that *the WZ coupling $S_{\text{WZ}}$ will not change the results of sections 2.1 and 2.2 because it will not modify the energy-momentum tensors of p-forms and/or extended objects*. This is the reason why we wrote footnote 4.

In the tadpole cancellation condition (40), only the first term in $S_{\text{WZ}}$ was considered, and so, the other (non-minimal) terms will modify that condition. We comment about it in our conclusions (see the top of page 16). Discussing the energy conditions validity for a general configuration of extended sources and their full coupling with all p-forms in the theory considered is beyond the scope of the paper, so we simply comment on how the fluxes will modify (42): there will be an extra term in equation (40) and (42), coming from higher order terms in $S_{\text{WZ}}$. We agree with the referee that the new flux contribution will appear as a *positive* term on the left-hand side of equation (40). Hence, it will induce a positive term in the last line of (41) and a negative one on the right-hand side of the inequality (42), without changing the inequality sign, i.e., (42) would be modified to
\begin{equation}
N_{\text{D}p}\geq 2^{p-5}(N_{\text{O}p_+}- N_{\text{O}p_{-}}) - N_{\text{flux}}/\mu_{\text{D}p}.
\end{equation}
In summary, the conclusions of sections 2.1 and 2.2 on the validity of energy conditions for p-forms and extended objects are not modified by the extra WZ flux couplings, and the conclusions in section (2.3) will be slightly modified because of extra terms in equations (39)-(42). The conclusion that the tadpole cancellation has to be considered for the validity of the energy conditions is unchanged, but the condition for positive total tension is modified if fluxes are included. We decided not to discuss these modifications further because (i) they are way beyond the scope of the paper and (ii) they are more suitable for a full analysis of more complex D and O-planes configurations.

We hope this addresses the referee's concern and our work can be published.

Sincerely,
Heliudson Bernardo (on behalf of all authors)

Anonymous on 2023-09-25  [id 4006]

(in reply to Heliudson Bernardo on 2023-09-07 [id 3960])

I thank the authors for their reply. It is perfectly fine that taking into account the fluxes is beyond the scope of the paper. However, the way the paper is presented right now let’s one think that it would be straightforward and would not change the results, which I highly doubt is true. I disagree with the statement “the conclusions in section 2.3 will be slightly modified because of the extra terms in equations (39)-(42)”, or at least the analysis of the present paper does not allow to conclude that. I would therefore like the authors to state clearly in the paper that they are not considering fluxes in the present work. By the way, when adding the last term due to the fluxes to the inequality written in their response, the latter becomes an equality.

---

## Round 4 · Author Response

Dear Editor

The authors thank the referee for reading our work and for the report. We reply to the referee's requested changes as follows:

Reply to request (1): We agree that fluxes will change the tadpole cancellation condition used in section 2.3, and recognize that in our conclusion section (see the last paragraph on page 15). However, the analysis in section 2.1 about the validity of the energy conditions by $p$-form fields is not changed because the extra Chen-Simons terms in the $p$-form action will not change their energy-momentum tensor. To highlight that, we wrote footnote 4.

Reply to request (2): By mean curvature, we meant precisely the left-hand side of the inequality (62). We admit that this requires extra clarification, so we slightly changed the text below (62). About the referee's intuition, since the WEC is satisfied for dS spacetime, we couldn't find any contradictions in our results.

Reply to request (3): We had the more recent progress in mind. We have rephrased the sentence as the referee suggested.

Reply to request (4): We meant essentially the SEC inheritance. We have rephrased the sentence for clarification.

Reply to request (5): We agree that taking a holistic view of the problem is our opinion. We have rephrased the sentence to make that explicit.

We thank the referee for spotting the typos. We have corrected them accordingly.

We hope this addresses all the referee's concerns and that our work can be published.

Sincerely,
Heliudson Bernardo (on behalf of the authors)

---

## Round 4 · List of Changes

1) Changed sentence below eq. (62);

2) The first sentence of the abstract was rewritten for clarification;

3) In the second to last paragraph of the introduction, the first sentence was rewritten for clarification;

4) In the first paragraph of section 4, the sentence starting in the first line was rewritten to express the authors' opinion better, as suggested by the referee.

(5) Few typos corrected.

---

## Round 5 · Author Response

Dear Editor,

We have added a footnote four on page 17 explaining how fluxes would affect our results. The comments are essentially a summary of the previous reply to the referee. We have also improved the discussion on DEC for p-forms at the end of section 2.1.

Sincerely,
Heliudson Bernardo

---

## Round 5 · List of Changes

-Improved discussion on DEC for p-form fields added;

-Footnote four modified to explain how fluxes would affect our results.

---

## Editorial Decision

published